# Unconventional fractional quantum Hall states and Wigner crystallization in suspended Corbino graphene

Manohar Kumar [1,2], Antti Laitinen[1] & Pertti Hakonen [1]

Competition between liquid and solid states in two-dimensional electron systems is an intriguing problem in condensed matter physics. We have investigated competing Wigner crystal and fractional quantum Hall (FQH) liquid phases in atomically thin suspended graphene devices in Corbino geometry. Low-temperature magnetoconductance and transconductance measurements along with *IV* characteristics all indicate strong charge density dependent modulation of electron transport. Our results show unconventional FQH phases which do not fit the standard Jain's series for conventional FQH states, instead they appear to originate from residual interactions of composite fermions in partially filled Landau levels. Also at very low charge density with filling factors $\nu \lesssim 1/5$, electrons crystallize into an ordered Wigner solid which eventually transforms into an incompressible Hall liquid at filling factors around $\nu \leq 1/7$. Building on the unique Corbino sample structure, our experiments pave the way for enhanced understanding of the ordered phases of interacting electrons.

[1] Low Temperature Laboratory, Department of Applied Physics, Aalto University, Espoo, Finland. [2] Laboratoire Pierre Aigrain, Département de Physique de l'École Normale Supérieure -PSL Research University, CNRS Université Pierre et Marie Curie-Sorbonne Universités, Université Paris Diderot-Sorbonne Paris Cité, Paris, France. Correspondence and requests for materials should be addressed to P.H. (email: pertti.hakonen@aalto.fi)

In the extreme quantum limit, the interplay between kinetic and potential energy of electrons in the two-dimensional electron gas leads to formation of exotic states like fractional quantum Hall (FQH) state, a many-body state where elementary excitations have fractional electronic charge[1], or even formation of electron solids, i.e., Wigner crystals where electrons freeze into a periodic lattice at very low temperature[2]. The FQH states are incompressible liquid states with quantized Hall conductance at distinct fractional fillings $\nu = n\phi_0/\mathbf{B}$, specified by integer values $\nu_{CF}$, $m = 1, 2\ldots$ according to $\nu = \nu_{CF}/(2m\nu_{CF} \pm 1)$, at charge density $n$ and magnetic field $\mathbf{B}$[3]. Here $\phi_0 = h/e$ is the flux quantum. In the presence of strong Coulomb interactions, electrons may eventually condense into quasiparticles of unconventional fractional states $\nu$ at which $\nu_{CF}$ attains non-integer values[4]. Some of these unconventional FQH states are of special interest due to the non-Abelian properties of their quasiparticles[5]. The manifestation of these exotic states requires a large Coulomb interaction energy compared with the disorder potential, putting strict requirements for the quality of the two-dimensional electron gas (2-DEG), the strength of the magnetic field, and for excitation by environmental noise or temperature.

Owing to reduced screening in atomically thin graphene, the electrons interact with higher Coulomb interaction energy than in conventional semiconductor heterostructures, providing an extraordinary setting for both unconventional FQH states and Wigner crystals. However, to our knowledge there are yet no previous experimental results showing manifestation of these exotic states in graphene. So far, most of the quantum Hall studies in graphene have been performed using supported samples (G/h-BN) in Hall bar geometry. The multi-terminal magnetotransport measurements upto 35 T on these samples have revealed several FQH states of mostly integer multiples of 1/3 (i.e., 1/3, 2/3…13/3)[6]. Contrary to this, despite of the higher Coulomb interaction energy in suspended graphene compared to supported samples and the very first observation of fractional state of $\nu = 1/3$ in rectangular two-terminal high-mobility suspended samples, not much progress has been seen in magnetotransport in suspended samples[7,8]. It was realized that these difficulties are mostly due to either intermixing of $\rho_{xx}$ and $\rho_{xy}$[9] or strain-induced pseudo-magnetic fields[10] in two-terminal and multi-terminal suspended graphene samples. This limits the experimental verification of fractional states to either local compressibility measurements using SET techniques[11] or transconductance measurements[12], both of which techniques have limited access to the complete electrical transport characteristics of fractional states; they probe local properties only. Despite of the experimental challenges, the higher Coulomb interaction energy in suspended graphene[13] makes it a very attractive platform to probe the interaction-based physics in conjunction with mechanical motion[14] and to investigate the competition between the lowest order FQH states and Wigner crystal-type of ordering[2].

In this work, we have fabricated suspended graphene Corbino disks for probing correlated many-body physics in graphene. The Corbino geometry outperforms the regular Hall bar geometry, as it directly probes the bulk two-dimensional electron gas (2-DEG) without having complications due to edge state transport[15]. In this geometry, the quantum Hall edge states counterpropagate at the perimeters of Corbino disk and no edge channel is formed between the two electrodes. Very recently, such quantum Hall states have been probed in graphene by magnetotransport measurement in Corbino geometry, also in on-substrate devices[16,17]. These measurements with on-substrate graphene Corbino disks have failed to show any fractional states, perhaps due to strong charge inhomogeneity induced by the substrate. Contrary to earlier approaches, in our experiment on suspended graphene Corbino disks, the quantum transport is dominated by activation

or tunneling across quantum states localized on static disorder potential. In suspended graphene, impurities and uncontrolled charge doping create a disorder potential forming localized states; which plays an important role in the magnetotransport. Like in quantum Hall states, in fractional quantum Hall regime, these states are formed to regions where the charge density does not lead to complete filling of a FQH state[18]. An incompressible FQH liquid surrounds the localized states, inside which free rearrangement of charge leads to a constant electrochemical potential. Such edge states can carry current across the sample only via coupling, either by quantum tunneling or by thermal activation, to adjacent localized states. Quasi-particles in single units can be added into the localized states, and thus such states behave quite like quantum dots in semiconducting devices. In the case of weak screening, these localized states interact electrostatically with many nearby localized states. Charge fluctuations among the interacting quantum dots lead to broadening of the energy levels, which enhances probability of quantum tunneling across localized states at low temperature. The charging effects lead to sets of modulation patterns in the current $I(\mathbf{B}, V_g)$ through the device as an AC gate voltage $V_g$ is applied. This is presented as transconductance maps $g_m(\mathbf{B}, V_g) = \mathrm{d}I/\mathrm{d}V_g$ which facilitates identification of localized states by pattern recognition and their classification according to the dominant filling factors[12].

In our magnetoconductance and transconductance measurements, we have resolved a distinctive set of incompressible liquid states with fractional filling factors of {−1/3, 1/5, 2/7, 4/13, 1/3, 4/11, 2/5, 3/7, 4/9, 4/7, 3/5, 2/3, 4/5, 4/3}. Of these FQH states, $\nu =$ 4/13 and 4/11 are unconventional states, formed due to interactions between composite fermions[19]. Our activation transport measurements on $\nu = 4/13$ and $\nu = 4/11$ states verify their incompressible nature with an energy gap of the order of 2% of the gap energy at $\nu = 1/3$. In addition to our magnetoconductance measurements, $IV$ characteristics and microwave spectroscopy showed evidence of solid phase order of electrons, i.e., Wigner crystallization, at low densities around $\nu \simeq 1/5 - 1/7$[20,21].

## Results

**Fractional quantum Hall states of composite fermions.** Our experiments address many-body electron states in a high-quality suspended Corbino disk (Fig. 1) at temperatures $T = 0.01 - 4$ K up to magnetic fields of 9 T. The fabrication of suspended Corbino disks and their characterization is described in Supplementary Notes 1–3. The basic characteristics of our Corbino samples are shown in Table 1. The magnetoconductance measurement represented as a Landau fan diagram clearly shows many basic integer Landau levels (2, 6, 10, 14) and broken symmetry states (1, 3, 4), as well as several conventional fractional states such as 1/3, 2/5, 3/7, 4/7, 3/5, and 2/3. However, to identify fractional states more distinctly, we performed transconductance analysis either

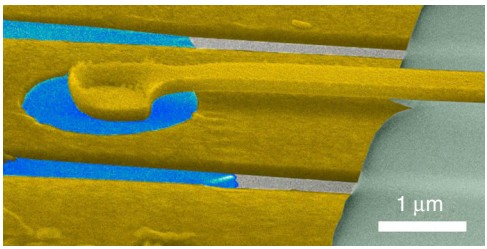

**Fig. 1** Suspended graphene Corbino disk. Scanning electron micrograph of a 2-µm-diameter graphene Corbino sample (XD) with 700 nm diameter circular middle contact; blue denotes the suspended graphene and gold/chromium contacts appear as grayish yellow. See the Supplementary Note 1 for detailed structure

**Table 1 Characteristics of our suspended Corbino graphene samples (by columns)**

| Sample | $r_i/r_o$ in μm | $n_0$ 1/cm² | $\mu_f$ cm²/Vs | $\sigma_0$ S | Measurement | FQH states |
|---|---|---|---|---|---|---|
| XD | 0.35/1.0 | $2 \times 10^{10}$ | $4.0 \times 10^4$ | $1.6 \times 10^{-5}$ | $g_m$ and $G_d$ | $\{\frac{1}{3}, [\frac{2}{5}], [\frac{4}{7}], \frac{2}{3}, [\frac{4}{5}], [\frac{4}{3}]\}$ |
| EV | 0.40/1.6 | $6 \times 10^9$ | $1.2 \times 10^5$ | $1.4 \times 10^{-4}$ | $g_m$ and $G_d$ | $\{[\frac{2}{7}], \frac{1}{3}, \frac{4}{11}, \frac{2}{5}, \frac{3}{7}, [\frac{4}{9}], \frac{4}{7}, \frac{3}{5}, \frac{2}{3}\}$ |
| EV* | 0.40/1.6 | $6 \times 10^9$ | $1.1 \times 10^5$ | $1.1 \times 10^{-4}$ | $G_d$ and $\delta G_d/\delta n$ | $\{\frac{1}{3}, \frac{2}{5}, \frac{3}{7}, \frac{3}{5}, \frac{2}{3}, \frac{4}{3}\}$ |
| EV₂ | 0.75/1.9 | $5 \times 10^9$ | $1.3 \times 10^5$ | $2.0 \times 10^{-4}$ | $G_d$ and $\delta G_d/\delta n$ | $\{\frac{1}{5}, \frac{2}{7}, \frac{4}{13}, \frac{1}{3}, \frac{4}{11}, \frac{2}{5}, \frac{3}{7}, \frac{4}{7}, \frac{3}{5}, \frac{2}{3}, \frac{4}{3}\}$ |

Inner and outer radii $r_i$ and $r_o$, respectively, $n_0$ is the residual charge density, field effect mobility $\mu_f$ is measured at $n \sim 5 \times 10^{10}$ cm⁻², and $\sigma_0$ denotes the minimum conductivity; the contact resistance $R_c$ was subtracted off from the conductivity only when calculating $\mu_f$. The FQH states probed by $g_m$ are indicated by square brackets $[p/q]$, otherwise the identification is based on differential magnetoconductance $G_d = dI/dV|_{V=0}$. The label EV* corresponds to a set of measurements done on sample EV in its second cool down, in which a change in FQH state quality is observed along with decreased mobility

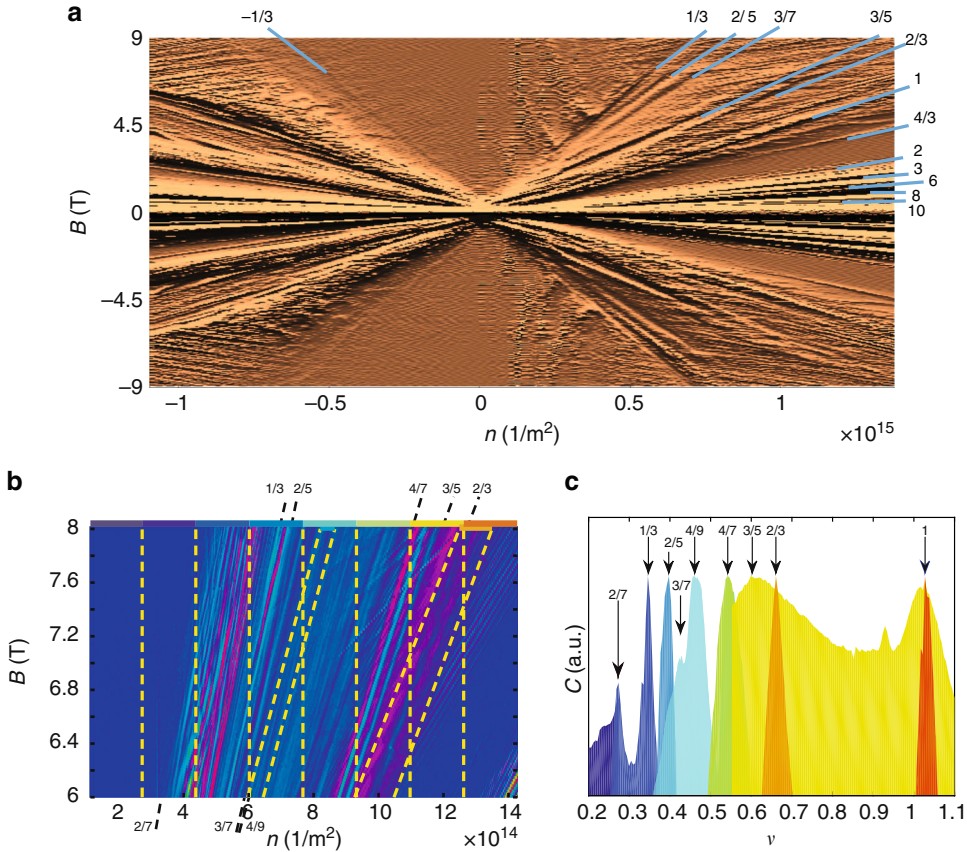

**Fig. 2** Magnetoconductance/transconductance measurement. **a** The derivative of logarithmic differential magnetoconductance w.r.t. $n$, i.e., $\delta \log(G_d)/\delta n$, of sample EV* (Table 1), displayed as a Landau fan diagram on the **B** vs. charge density $n$ plane. The blue lines at $\nu = nh/e\mathbf{B}$ identifying regular QH states $\nu = \{2, 6, 10\}$, broken symmetry states $\nu = \{0, 1, 3\}$ originating from lifting spin and valley degeneracies, and FQH states $\nu = \{-1/3, 1/3, 2/5, 3/7, 3/5, 2/3, 4/3\}$. **b** Modulation of transconductance phase $\mathrm{Arg}\{g_m\}$ measured for sample EV over charge density $n$ ($V_g$ up to 15 V) at magnetic fields between 6 and 8 T (filling factor $\nu = 1$ is reached at the lowest right corner). The data are divided into ten regions, each marked with a color bar at the top of the stripe: we use eight vertical stripes of equal size and two inclined stripes which are defined in order to improve the contrast of fringes at $\nu = 3/7$ and $\nu = 2/3$. The correlation analysis has been performed separately in each region. The correlation function $h(\nu)$ specifies the structure of these fringes w.r.t. the slope $\delta n/\delta \mathbf{B}$ equaling to the filling factor $\nu$. The black dashed lines off the borders indicate FQH states with the marked fractional filling factor. **c** The scaled cross-correlation function $c(\nu) = h(\nu)/h(\nu)_{\max}$, for all values of $\nu = 0.2...1.1$, identifying fractional fillings $\{2/7, 1/3, 2/5, 3/7, 4/9, 4/7, 3/5, 2/3\}$, in agreement with our magnetoconductance measurements; see the Supplementary Note 4 for further analysis. The color of the distribution refers to the color code of the analyzed regions in Fig. 2b. The peak values are marked with the corresponding fractional filling factors. Further FQH states are displayed in inverse differential conductance data in Fig. 3 (see also Supplementary Note 5). The measurement was performed at mixing chamber temperature of 20 mK

based on numerical derivative of logarithmic differential conductance $\delta \log G_d/\delta n$ (Fig. 2a), or on lockin measurements of $g_m(\mathbf{B}, V_g)$ (Fig. 2b). Both FQH and QH states, localized on disorder potential landscape, are observed as additional sets of parallel lines in our transconductance scans. The slope of the fringes varies in non-monotonic fashion, which is indicative of the

features coming from spatially modulated charge density due to disorder potential and corresponding to filling factor $\nu$. These fringes have been treated using a cross-correlation analysis described in the Supplementary Note 4.

The majority of these fractional states in two-dimensional electron gases can be described using Laughlin's wave function

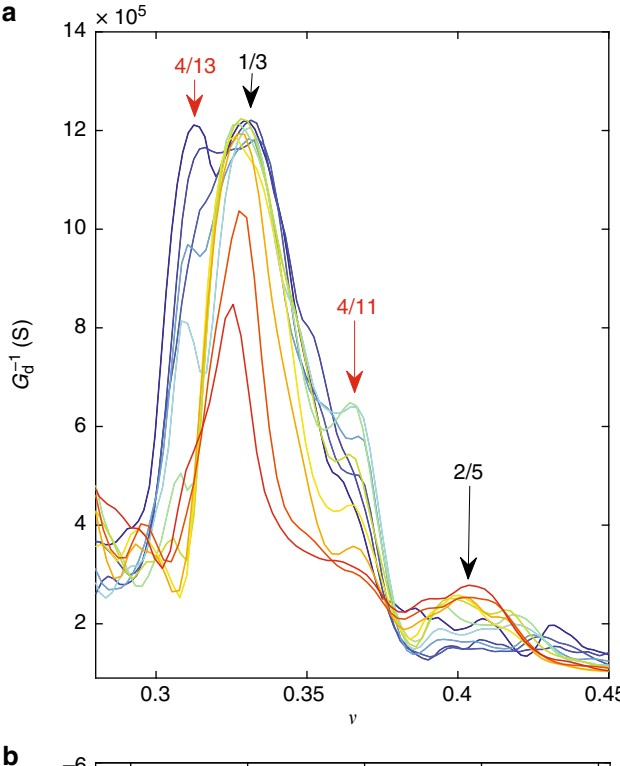

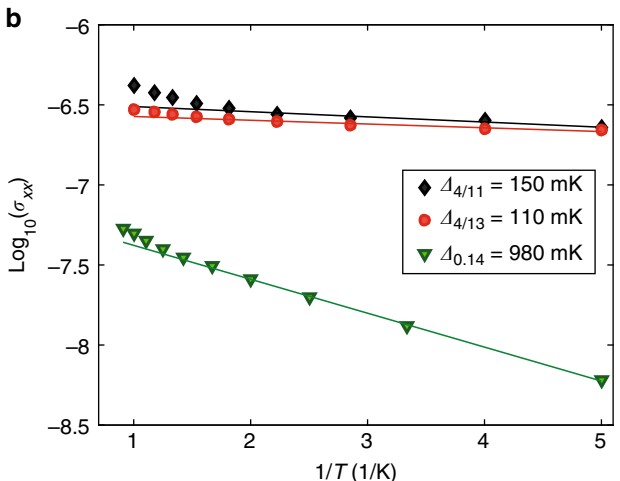

**Fig. 3** Unconventional FQH states. **a** Inverse differential conductance through the Corbino disk measured at AC for sample EV2 at magnetic fields between 6 and 8.5 T, measured at 20 mK. The magnetic field varies from 8.5 T (blue) to 6 T (red). The fractional states are identified by peaks at $\nu = nh/e\mathbf{B}$ in inverse conductance traces which collapses on top of each other in $G_d^{-1}$ vs. $\nu$ plot. Distinct states at $\nu = \{4/13, 1/3, 4/11, 2/5\}$ are identified from measured traces. **b** The Arrhenius plot of DC conductivity $\sigma_{xx}$ at $\mathbf{B} = 7$ T for fractional states at $\nu = 4/11$ (black diamonds) and $\nu = 4/13$ (red circles), contrasted to an activation measurement at $\nu = 0.14$ (green inverted triangles) at $\mathbf{B} = 9$ T. The straight lines represent Arrhenius fits that were used to extract the gaps of these states. The obtained gaps are also displayed as an inset

and Jain's composite fermion (CF) theory. In the CF picture, fractional states are interpreted as integer quantum Hall effect of flux-transformed noninteracting composite particles. These CFs are quasiparticles of an electron bound to an even number of flux quanta in a effective magnetic field $\mathbf{B}^\star = \mathbf{B} - 2\phi_0 n$. Even though Jain's theory of CF considers these particles as noninteracting entities, there may remain small interactions between the CFs which lead to the formation of exotic states, e.g., fractional states

of CFs[19]. These unconventional fractional quantum Hall states reside at fillings $\nu^\star_{CF} = \nu_{CF}/(2m\nu_{CF} \pm 1)$, where $\nu_{CF}$ has fractional non-integer values and the integer $m$ is the number of flux quanta[22]. Evidence of such interacting CF states has been obtained by Pan et al. in GaAs/AlGaAs[4]. Despite of the stronger electron–electron Coulomb interactions, these unconventional fractional states of composite fermions have remained elusive in graphene. Our present data provides the evidence of unconventional FQH states $\nu = \{4/13, 4/11\}$ in suspended graphene.

These states are distinctly seen as peaks in $G_d^{-1}$ at filling factors across $\nu = 1/3$, as depicted in Fig. 3a. The $G_d^{-1}$ vs. $\nu$ plot for fields $\mathbf{B} = 6$–8.5 T shows distinct peaks at $\nu = 4/13$ and $4/11$. Since these peaks seem to evolve at the specific filling factors without significant movement, we conclude that they are due to unconventional fractional states rather than replicas of other conventional FQH states shifted by local charge inhomogeneity. The $\nu = 4/11$ corresponds to the fractional filling $\nu_{CF} = 1 + 1/3$ for two flux quanta composite fermion ($m = 1$), and $\nu = 4/13$ corresponds to $\nu_{CF} = 1 + 1/3$ for four flux quanta composite fermion ($m = 2$)[19,23,24].

In the temperature dependence of magnetoconductance, the incompressibility of the FQH states shows up as an activated behavior in the DC conductivity $\sigma_{xx} = \sigma_0 \exp(-\Delta_\nu/2k_BT)$. From the Arrhenius plot shown in Fig. 3B, the linear fit to $\log_{10}\sigma_{xx}$ vs. $1/T$ yields the activation energy $\Delta_\nu$. The activation energy of $\nu = 1/3$ state amounts to $\Delta_{1/3} = 4.7$ K at 7 T, in agreement with Ghahari et al.[25] Similarly, the activation energies for $\nu = 4/13$: $\Delta_{4/13} = 110$ mK and $\nu = 4/11$: $\Delta_{4/11} = 150$ mK. These energies are about 2% of the activation energy of the $\nu = 1/3$ energy gap, in agreement with theoretical expectations[22].

**Wigner crystallization**. The FQH states at the lowest charge densities are of special interest; the correlated liquid states are expected to be terminated in an electron solid phase. The formation of this electron solid, i.e., the Wigner crystal, is related to two competing energy scales: the Coulomb potential energy and the kinetic energy. At a sufficiently low charge density and high magnetic field, the Coulomb energy dominates and, hence, electrons crystallize into a two-dimensional electron solid. In our experiments around $\nu = 1/7$–1/5, we find evidence for such incompressible solid states within the lowest Landau level. The Landau fan diagram for $\nu < 1/5$ is shown in Supplementary Note 5. We affirm the incompressibility of the region ($\nu = 1/7$–1/5) by activation transport, non-linear current-voltage characteristics, and microwave spectroscopy (for background, see Supplementary Notes 6–9).

Upon the onset of Wigner crystallization, electrons undergo a phase transition at which structural order and extent of correlations are enhanced. Thermodynamically, the incompressibility of a system scales up as it undergoes a phase transition from liquid to solid. We probed this increased incompressibility of electrons in the Wigner solid phase by performing microwave absorption spectroscopy. The measurement scheme is shown in the Supplementary Note 9. An rf signal of amplitude −68 dBm was fed from the microwave generator to the inner contact, and the DC conductance was simultaneously measured at a 50 μV bias voltage using a transimpedance amplifier. For the rf-power of −68 dBm, the mixing chamber temperature stayed within $20 \pm 10$ mK. However, assuming validity of the Wiedmann–Franz law, the minimum achievable electron temperature at this rf-power remains around 0.4 K according to the regular hot electron model.

Figure 4a displays the measured DC current $I$ while sweeping the frequency of the rf signal over the range 0.01–5 GHz. The data in Fig. 4a display a strong resonance at $f_P = 3.0$ GHz for $\nu = 0.16$.

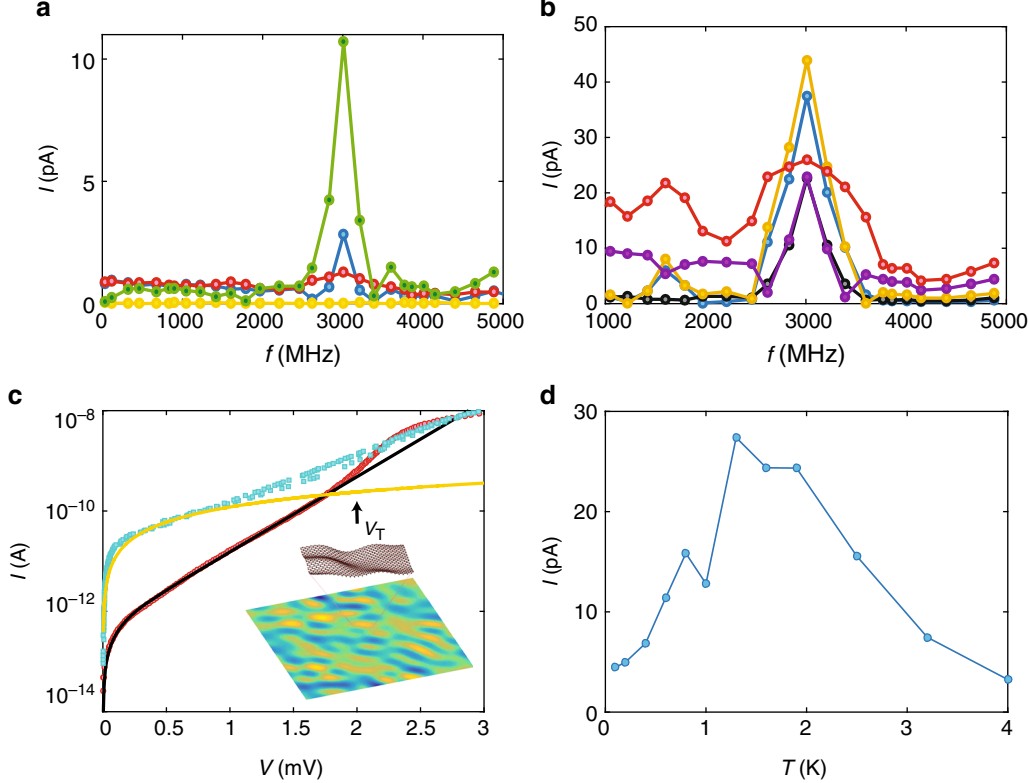

**Fig. 4** Wigner crystal. **a** The absorption microwave spectra of the detected absolute value of the DC current $I$ vs. irradiation frequency $f$ for sample EV2, measured at $\nu = 1/3$ (red), $\nu = 0.16$ (green), $\nu = 0.15$ (blue), and $\nu = 0.12$ (yellow), and at magnetic field $\mathbf{B} = 9$ T and rf-power $= -68$ dBm. An absorption resonance with a maximum response for $\nu = 0.16$, is seen at frequency 3 GHz which is identified as a collective mode frequency in the disordered-pinned Wigner crystal phase (see text). The flat curve at $\nu = 0.12$ is an indication of a strongly gapped fractional quantum Hall state. **b** Microwave absorption spectra in terms of DC current $I$ measured across the $\nu = 0.16$ (at fixed charge density, $V_g = 2.8$ V, rf-power $= -65$ dBm, $\mathbf{B} = \{9$ T (blue), 8.8 T (black), 8.6 T (yellow), 8.4 T (violet), 8 T (red)$\}$). Broadening of the resonance peak is seen clearly at $\mathbf{B} = 8$ T, because the WC is closest to its melting temperature at this field. Note the increase in the DC current compared to Fig. 4a (see also Supplementary Note 9). **c** Semilog plot of two $IV$ curves from sample XD corresponding to filling factors in frame **a**: (red circles) $\nu = 0.16$ and (blue squares) $\nu = 0.33$. The black curve denotes a fit of $I_W$ (Eq. 1) describing electrical transport in the Wigner crystal regime by thermally activated depinning of crystallites, while the orange curve displays linear $IV$ behavior with a total resistance of 8 MΩ due to quantum tunneling over a chain of localized states. This is clearly different from Wigner crystal behavior and well valid up to 1 mV. $V_T$ marks the threshold voltage. The inset displays a disordered Wigner crystal where charge ordering follows corrugations of suspended graphene: higher charge density (yellow) resides at the dimples. **d** The temperature dependence of the microwave absorption peak of Fig. 4a at the Wigner crystal filling factor $\nu = 0.15$. The maximum in the current indicates the melting point of the Wigner crystal order

This resonance is visible in the region $1/7 < \nu < 1/5$, while it becomes very weak at $\nu = 1/3$ and fully absent at $\nu \leq 0.12$. This resonance in microwave absorption spectra, observable up to $T \sim 1.4$ K, can be understood either as a resonance of individual localized particles or as a collective resonance of an incompressible, ordered phase. For photon-induced carrier excitation at $f_P = 3$ GHz from localized states, the critical temperature for thermally induced single carriers would be $T_I < hf_P/k_B \sim 140$ mK. Contrary to this, the absorption resonance is observed up to $T_m = 1.5$ K, exceeding $T_I$ by an order of magnitude. At such a high temperature, the single particle localization as formulated above is not effective and all particles would become delocalized. Hence, this peak in the microwave absorption spectra points towards the existence of an incompressible solid as for its origin. In the presence of an external oscillating field, the electrons in the Wigner crystal domains oscillate within their pinning potential with a collective resonance frequency of $f_P$[26]. Also, for the correlated Laughlin liquid, an rf-field will excite the lowest lying collective excitation, the magnetoplasmon of the FQH state. The energy scale of magnetoplasmons ($E/h \sim 100$ GHz for our sample size) is much larger than that of transverse phonon-like modes in the Wigner

crystal and, consequently, these plasmons remain beyond reach in our experimental setting[26–28].

The sharpness of the low-temperature resonance spectra is at odds with the presence of complex non-uniform states, such as Wigner glass or non-homogeneous two-fluid states, which would tend to give a broader microwave resonance signal. The microwave absorption spectroscopy was also performed at two times larger rf-irradiation power ($P = -65$ dBm) in the Wigner crystal regime ($0.15 < \nu < 0.17$) (Fig. 4b). The resonance at $\nu = 0.16$ is the sharpest, while away from $\nu = 0.16$, the resonances are either broader or of smaller amplitude. At the rf-power of $-65$ dBm, the combined action of increased activated transport and enhanced Joule heating due to rf absorption leads to an increase in the DC current $I$, see also Supplementary Note 9.

For further proof of the Wigner crystal, we also probed the $IV$ characteristics of our sample. The non-linear electrical conductance provides a means to distinguish between several possible transport mechanisms, including activation of fractionally charged quasiparticles[29] and depinning of Wigner crystal[30,31]. The $IV$ characteristics and the relevant transport mechanisms at low-filling factor ($\nu = 0.16…0.33$) are discussed in the Supplementary Note 7 for sample EV. The behavior of our non-linear $IV$ characteristics for sample XD is illustrated in

Fig. 4c on a semilog scale. The data at $v = 0.16$ display exponential increase in the current $I$ between $V = 0.1$–1.8 mV. Such exponential increase in current has been identified as gradual depinning of the ordered charge carrier phase, having either charge density wave (CDW) or Wigner solid-type of order; it is more likely to be a Wigner crystal order at low charge density in graphene[32,33]. Deviation of exponential behavior starts at $V_T = 1.8$ mV which is identified as the depinning threshold of Wigner crystal sliding. When $V < V_T$, charge transport takes place by thermally activated hopping of pinned regions. Similar behavior was observed in sample EV2, where we obtained a threshold voltage of $V_T = 1.4$ mV.

Following ref. [31], we write for the current due to thermally activated motion of pinned crystallites below $V_T$ as

$$I_W = e^* f_a \left\{ \exp\left[ -\frac{\bar{\Delta} - eV/2N}{k_B T} \right] - \exp\left[ -\frac{\bar{\Delta} + eV/2N}{k_B T} \right] \right\} \quad (1)$$

where $e^*$ is the effective charge, $f_a$ is the depinning attempt frequency, $\bar{\Delta}$ is the average value of the pinning potential, and $N$ denotes the number of Wigner crystallites in series across the transport path. In Fig. 4c, we obtain a good agreement between this basic theory and our experimental data at $v = 0.16$ using parameters $e^* = e$, $f_a = 3.0$ GHz, $\bar{\Delta} = 140$ µeV, $N = 7$, and $T = 0.2$ K. As expected for $N$ Wigner crystallites in series, $N\bar{\Delta}/e \simeq V_T/2 = 1.0$ mV for XD.

To elucidate the strength of the pinning barrier, we probed the melting temperature of the Wigner crystal by recording the amplitude of the pinning resonance of Fig. 4a as a function of $T$, which is presented in Fig. 4d. The increase in the electronic temperature due to the −68 dBm rf-excitation is much smaller than the melting temperature, but it may account for the saturation in $\Delta I$ at the lowest temperatures. Hence, the initial rise in the peak current at low temperatures is due to heating assisted activated transport upon rf-irradiation, while at higher temperature, melting of the Wigner crystal results in a weak temperature dependence of $I(T)$ leading to a decrease in the peak amplitude[34]. Figure 4d, shows $T_m \sim 1.5$ K, which agrees closely with the fitted activation barrier $\bar{\Delta} = 170$ µeV. Also, this $T_m$ correlates with the classical theoretical estimate of the melting temperature $T_m = e^2 \sqrt{n} \left( 4\pi\varepsilon_0 \varepsilon_g k_B \Gamma \right) \simeq 1$ K[35], where $\varepsilon_g = 3$, is the dielectric constant for graphene and $\Gamma = 130$[36]. Besides the non-quantum formula for shear modulus, this comparison is influenced by uncertainty due to non-uniformity and irregular melting of the Wigner crystal. Clearly, our microwave resonance data in Fig. 4b indicate the presence of crystallites which melt before the electronic temperature has reached the dominant $T_m$ shown in Fig. 4d.

In our rf-spectroscopy measurements, we investigated microwave-induced phenomena also around the pinning frequency $f_p = 3.0$ GHz. In general, a coherent crystallite averages over the pinning potential and the frequency becomes smaller with enhanced crystal quality. Consequently, an unavoidable spread in crystallite frequencies exists and it is expected to lead to a broad resonance, which is observed in our experiments under large rf-irradiation as illustrated in Fig. 4b. We can estimate the average domain size of pinned Wigner crystallites $L_c = 0.70$ µm from classical-shear modulus $c$ and resonance frequency $f_p$[37] (Supplementary Note 8). In comparison to thermal depinning model, the size of the crystallites obtained here is significantly larger. The discrepancy could be due to an overestimation of $L_c$ by the classical-shear-modulus model where the renormalization of the traverse phonon dispersion relation by quantum effects has been ignored, although this is disputable at very low charge density and in high magnetic field[38]. Additionally, in thermally

activated transport near the threshold voltage, the Lorentz force will become comparable to the electric field and hence, the so-called gyrotropic tunneling of quasiparticles[39] will influence the sliding of the Wigner crystal. For example, by using a 45° sliding angle w.r.t. to radial direction, a perfect fit using $N = 3$, $T = 0.4$ K is achieved for our experimental non-linear $IV$ data, which removes the discrepancy between the microwave absorption and $IV$ measurements. Nevertheless, the average domain size $L_c > 3 a$, where $a \sim 1/\sqrt{\pi n}$ is the Wigner crystal lattice constant, verifies that well-defined local crystalline order persists in our sample.

One factor enhancing the observability of the Wigner crystal in suspended graphene may be provided by the deformation of the graphene sheet into a dimple lattice under the action of the back-gate voltage (see the inset of Fig. 4c). This would add inertia to the charge carriers, as is the case with electrons on superfluid helium where the Wigner crystal has been observed even in the absence of magnetic field[40]. Frozen-in dimples, on the other hand, may act as important pinning sites, in addition to the charged pinning centers provided by residual charge density.

Our experiment clearly supports the presence of crystalline Wigner solid order in region $1/7 < v < 1/5$, whereas a strongly gapped state is observed at $v \ll 1/7$. This gapped state around $v = 0$ may be related to excitonic ordering. The crossover between these two regimes appears to take place at $v \simeq 0.14$, around which we find activated behavior with a gap of $\Delta_{v=0.14} \simeq 1$ K (Fig. 3b). Re-entrant behavior of the FQH order is one possibility for the observed behavior, but our experiment does not resolve any clear characteristics associated with a particular fractional state.

## Discussion

In suspended graphene, impurities and the back-gate-modified dimple lattice create disorder potentials which are screened by charge, leading to a constant electrochemical potential. But at very low charge density, the effectiveness of the charge screening breaks down, exposing hills and valleys of the disorder potential. The electrons will be localized in the valleys of the potential landscape, and they become rearranged in reduced dimension via Coulomb interactions. Owing to the reduced screening and dimension, the charges will interact strongly. In an intermediate regime where the kinetic energy is still appreciable, the composite fermions will condense to fractional states such as $v = 4/13$ and $v = 4/11$ in partially filled Landau levels. But at even lower charge density, the Coulomb interaction energy will strongly dominate over the kinetic energy, leading to freezing of the electronic states into more ordered states such as the Wigner crystal pinned by the potential landscape. Hence, the formation of exotic FQHE and WC states do both owe to the reduced dimensionality and higher Coulomb interactions in suspended graphene where, besides, the lattice of dimples could be a significant contributing factor. The experiment demonstrated here opens up the possibility for probing unexplored strongly correlated electron states which may emerge more markedly in suspended graphene than in any other system.

In conclusion, we have investigated fractional quantum Hall ordering in two-terminal suspended graphene Corbino devices where no edge states carry current across the sample. For bulk transport in the Corbino geometry, our measurements present an unparalleled sequence of FQH states in transport measurements on two-lead graphene. Besides many regular FQH states, we also resolved unconventional fractional quantum Hall states at filling factors $v = 4/13$ and $v = 4/11$. The presence of these states indicates residual interaction between composite fermions. At low-filling factors, we observe competition between liquid and solid

ordering. Our results, most notably microwave absorbtion spectra, support Wigner crystallization of electrons in region $1/7 < \nu < 1/5$ with a melting temperature of 1.5 K. The crossover from the strongly gapped quantum Hall state at $\nu = 0$ to the Wigner crystal regime, is found to involve an intermediate state with reduced incompressibility. Being a truly 2D system, graphene provides new possibilities for studies of ordered electron gas/liquid/solid phases with the added benefit that the structures exist right at the surface, fully exposed for probing.

## Methods

**Sample fabrication.** Our sample fabrication is based on well-chosen combination of resists with differential selectivity, which allowed the fabrication steps for deposition of a suspended top contact (see Supplementary Note 1 for details). We exfoliated graphene (Graphenium form NGS Naturgraphit GmbH) using a heat-assisted exfoliation technique to maximize the size of the exfoliated flakes[41]. Monolayer graphene flakes were located by their contrast in an optical microscope and verified using a Raman spectrometer with He–Ne laser (633 nm). The first contact defining the outer rim of the Corbino disk (Fig. 1) was deposited in the usual manner[42], and later the inner contact was fabricated along with an air bridge to connect the inner contact to a bonding pad (Supplementary Note 1). Strongly doped silicon Si++ substrate with 285 nm of thermally grown SiO$_2$ was used as a global back gate.

Due to the exposure of graphene to resists during the fabrication process, our suspended devices tend to be highly doped (mostly p-type) in our first resistance $R_d = dV/dI$ vs. gate voltage $V_g$ scans. Annealing of samples on LOR was typically performed at a bias voltage of $1.6 \pm 0.1$ V which is quite comparable with our HF etched, rectangular two-lead samples[43]. For further details on annealing, we refer to the Supplementary Note 2.

**Measurement techniques.** Our measurements down to 20 mK were performed in a BlueFors LD-400 dilution refrigerator. The measurement lines were twisted pair phosphor-bronze wires supplemented by three stage $RC$ filters with a nominal cut-off given by $R = 100$ Ω and $C = 5$ nF. However, due high impedance of the quantum Hall samples the actual cut-off is determined by the sample resistance. For magnetoconductance measurement, we used an AC peak-to-peak current excitation of 0.1 nA at $f = 3.333$ Hz.

For transconductance $g_m = dI/dV_g$ we measured both magnitude Mag$\{g_m\}$ and phase Arg$\{g_m\}$ using low-frequency lockin detection. The best results for $g_m$ correlation analysis were obtained by recording Arg$\{g_m\}$ at a bias voltage $V$ that corresponded to the onset of the $V^\alpha$ regime, see Supplementary Note 7. Consistent information was obtained by analyzing the simultaneously acquired Mag$\{g_m\}$. The gate frequency in AC transconductance measurements was set at $f = 17.777$ Hz, while the peak-to-peak AC excitation amplitude was adjusted to correspond to charge of one electron over the sample. The DC bias between source and drain was varied in the range $V = 0.1$–0.5 mV. Note that this technique was mainly applied to the Jain sequence states as the unconventional states ($\nu = 4/13$ and $\nu = 4/11$) were easily overshadowed by the close dominant $\nu = 1/3$ state in the cross-correlation analysis due to the nature of the technique, for further information see the Supplementary Note 4. We could also verify the unconventional fractional states in mechanical resonance measurements, see Supplementary Note 10.

**Data availability.** The data that support the findings of this study are available from the corresponding author upon reasonable request.

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

## Acknowledgements

We thank C. Flindt, A. Harju, Y. Meir, T. Ojanen, S. Paraoanu, E. Sonin, and G. Féve for fruitful discussions. This work has been supported in part by the EU Framework Programme (FP7 and H2020 Graphene Flagship), by ERC (Grant No. 670743), and by the Academy of Finland (Projects No. 250280 LTQ CoE and 286098). A.L. is grateful to Vaisälä Foundation of the Finnish Academy of Science and Letters for a scholarship. This research project utilized the Aalto University Otanano/LTL infrastructure.

## Author contributions

The research was initiated by P.J.H. The experimental principle was selected by M.K. and P.J.H. The sample structure was developed and manufactured by A.L. The experiments were carried out by A.L. and M.K. who were also responsible for the data analysis. The results and their interpretation were discussed among all the authors. The paper and its supplement were written by the authors together.

## Additional information

**Competing interests:** The authors declare no competing interests.

