## [Peer Review File · Nature Communications]

Reviewers' comments:

Reviewer #1 (Remarks to the Author):

I reviewed this manuscript in the previous submission, and I found that the manuscript is technically sound. I support the manuscript to be published in Nature Communication.

Reviewer #2 (Remarks to the Author):

In this work, transport measurements of a suspended graphene Corbino geometry structure in the quantum Hall regime is investigated. The speciality of the Corbino disc is that the edge states do not connect the source and drain leads but circulate at the boundary between the leads and the sample. The trans- and magnetoconductance measurements in this work reveal a wealth of fractional quantum Hall states not seen in earlier graphene Corbino disc geometries (which were not suspended). Presumably strong unscreened Coulomb interaction even formed exotic fractional quantum Hall (FQH) states beyond the composite fermion picture. Signs of such fractions could be detected, for the filling factors $4/13$ and $4/11$, in transport for the first time in suspended graphene (according to the authors).

At low filling factors (between $1/7$ and $1/5$) evidence of a Wigner crystal phase has been detected with several methods (microwave absorption, IV-characteristics) and compared to model calculations.

The work seems to me very complete (there is also extensive supplemental materials), mostly consistent and nicely presented. The question is how much new insights into graphene physics or support for possible applications does this work provide? It seems to me that the observed effects and the detected correlated phases have been seen in experiments (and were theoretically predicted) before, however, not all of these features in the same experiment. I think that the Corbino geometry has clear advantages over the conventional Hall bar structures when using the applied methods since the transport does not probe directly the edge states in this geometry and the signatures of the correlated bulk states and disorder pinnings are better seen in the Corbino geometry structure. I therefore think that the present experiment is an important step towards the detection of correlated states in graphene.

I have a few questions I would like the authors to consider:

1. The exotic FQH states for filling factors $4/13$ and $4/11$ seem not visible in the transconductance (Fig. 2), although many other fractions are observed there. What is the reason?

2. The microwave resonance associated with the collective pinning mode of the crystal (at ~ 3.0 GHz, see Fig. 4) can be distinguished from the plasmonic modes of the FQH states according to the authors due to different frequency ranges (see page 8). What are the typical numbers of plasmon excitation frequencies of the FQH states?

3. Fig. 3A: In the text (page 5) it says, the magnetoconductance is shown in Fig. 3A, but in this figure, R_d is shown (probably this means magnetoresistance). Also, the mentioned "dips" in the main text are "peaks" in Fig. 3A.

4. On page 4, the transconductance is introduced as $g_m(B,n)=dI/dV_g$, but the voltage V_g is not defined in the text.

Response to reviewer

Reviewer #1 (Remarks to the Author):

I reviewed this manuscript in the previous submission, and I found that the manuscript is technically sound. I support the manuscript to be published in Nature Communication.

We thank the Referee for supporting our work.

Reviewer #2 (Remarks to the Author):

In this work, transport measurements of a suspended graphene Corbino geometry structure in the quantum Hall regime is investigated. The speciality of the Corbino disc is that the edge states do not connect the source and drain leads but circulate at the boundary between the leads and the sample. The trans- and magnetoconductance measurements in this work reveal a wealth of fractional quantum Hall states not seen in earlier graphene Corbino disc geometries (which were not suspended). Presumably strong unscreened Coulomb interaction even formed exotic fractional quantum Hall (FQH) states beyond the composite fermion picture. Signs of such fractions could be detected, for the filling factors $4/13$ and $4/11$, in transport for the first time in suspended graphene (according to the authors).

At low filling factors (between $1/7$ and $1/5$) evidence of a Wigner crystal phase has been detected with several methods (microwave absorption, IV-characteristics) and compared to model calculations.

The work seems to me very complete (there is also extensive supplemental materials), mostly consistent and nicely presented. The question is how much new insights into graphene physics or support for possible applications does this work provide? It seems to me that the observed effects and the detected correlated phases have been seen in experiments (and were theoretically predicted) before, however, not all of these features in the same experiment. I think that the Corbino geometry has clear advantages over the conventional Hall bar structures when using the applied methods since the transport does not probe directly the edge states in this geometry and the signatures of the correlated bulk states and disorder pinnings are better seen in the Corbino geometry structure. I therefore think that the present experiment is an important step towards the detection of correlated states in graphene.

We thank the Referee for reviewing our work. We would like to note that while the subjects (Wigner crystallization and unconventional FQHE) considered here may have been observed in other 2DEG systems previously, they most definitely have not been reported in the context of graphene. Due to the special characteristics of graphene it is not obvious that everything happens the same way in it. Thus, our work provides unique insight into these intriguing, current topics.

I have a few questions I would like the authors to consider:

1. The exotic FQH states for filling factors $4/13$ and $4/11$ seem not visible in the transconductance (Fig. 2), although many other fractions are observed there. What is the reason?

The transconductance data was analyzed using the cross-correlation technique (reviewed in the Supplementary) where the states are identified by calculating sums of correlations along all lines in the mapping. This technique was employed in order to provide a systematic way of identifying the fractional states. However, the drawback of this technique is that small features cannot be distinguished if there are big features close by giving excess correlation, which is a natural consequence of the definition of our correlation function. This difficulty was realized from the beginning, for example in the case of the state $2/3$, where the problem was mitigated by tilting the area in which the correlations were calculated. In the case of $4/11$ and $4/13$, such an improved analysis did not work properly due to the dominance of the $1/3$ state in close proximity.

In the case of bare conductance, the observations were interpreted just by looking at the conductance traces, where the strong $1/3$ feature was not so problematic. Additionally, the large inverse conductance peak at $1/3$ in Fig. 3A was attenuated due to the sample-dependent RC time constant, which made the weak $4/13$ and $4/11$ features more discernible in this picture.

In order to communicate this to readers, we have added a sentence for clarification of this issue at the end of the Methods section in the manuscript.

2. The microwave resonance associated with the collective pinning mode of the crystal (at ~ 3.0 GHz, see Fig. 4) can be distinguished from the plasmonic modes of the FQH states according to the authors due to different frequency ranges (see page 8). What are the typical numbers of plasmon excitation frequencies of the FQH states?

At charge carrier densities relevant here, we have $\omega_p \sim 100$ GHz by using Eq. 3.8 from Ref. 26. Clearly this is beyond our measurement range. Nonetheless, we added this detail to the manuscript for clarity.

3. Fig. 3A: In the text (page 5) it says, the magnetoconductance is shown in Fig. 3A, but in this figure, R_d is shown (probably this means magnetoresistance). Also, the mentioned "dips" in the main text are "peaks" in Fig. 3A.

These are unfortunate typos - sorry for the confusion. The measurement inherently probes conductance. Fig. 3A presents the inverse differential conductance G_d^{-1} for clarity, and consequently the text and the caption should say "peaks". This has been corrected in the present version.

4. On page 4, the transconductance is introduced as $g_m(B,n)=dI/dV_g$, but the voltage V_g is not defined in the text.

This has now been clarified in the text.

REVIEWERS' COMMENTS:

Reviewer #2 (Remarks to the Author):

This is my second report of the paper "Unconventional fractional quantum Hall states and Wigner crystallization in suspended Corbino graphene disk". The authors have satisfactorily countered my criticism and amended the manuscript correspondingly. I therefore recommend the paper for publication in nature communication in the present form.

Response to reviewer

Reviewer #2 (Remarks to the Author):

This is my second report of the paper "Unconventional fractional quantum Hall states and Wigner crystallization in suspended Corbino graphene disk". The authors have satisfactorily countered my criticism and amended the manuscript correspondingly. I therefore recommend the paper for publication in nature communication in the present form.

We thank the Referee for supporting our work.